

# Thermalization of the quantum planar rotor with external potential

Birthe Schrinski[1], Yoon Jun Chan[1,2] and Björn Schrinski[1,3]

**1** Faculty of Physics, University of Duisburg-Essen, Lotharstraße 1, 47048 Duisburg, Germany
**2** Institute of Mathematics, Augsburg University,
Universitätsstr. 14, 86159 Augsburg, Germany
**3** Niels Bohr Institute, University of Copenhagen,
Blegdamsvej 17, 2100 Copenhagen, Denmark

## Abstract

We study decoherence, diffusion, friction, and how they thermalize a planar rotor in the presence of an external potential. Representing the quantum master equation in terms of auxiliary Wigner functions in periodic phase space not only illustrates the thermalization process in a concise way, but also allows for an efficient numerical evaluation of the open quantum dynamics and its approximate analytical description. In particular, we analytically and numerically verify the existence of a steady state that, in the high-temperature regime, closely approximates a Gibbs state. We also derive the proper classical limit of the planar rotor time evolution and present exemplary numerical studies to verify our results.

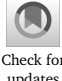

## Contents

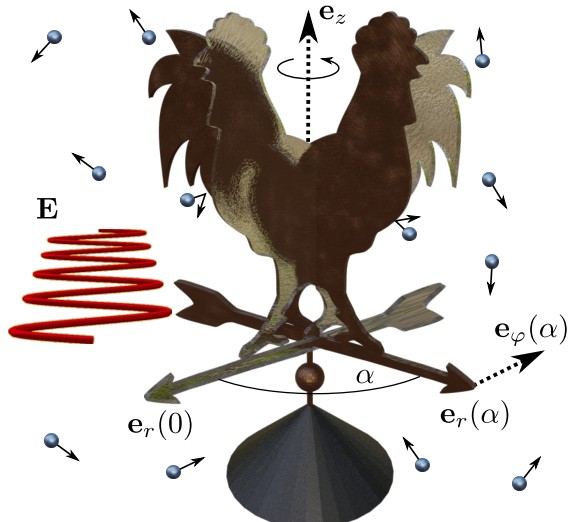

Figure 1: We study the time evolution of an arbitrary planar rotor revolving around $\mathbf{e}_z$ with the usual cylindrical coordinate vectors $\mathbf{e}_r(\alpha)$ and $\mathbf{e}_\varphi(\alpha)$ as function of the single degree of freedom $\alpha$. The quantum weathercock is coupled to a bath of temperature $T$, for example a gaseous environment. In addition, an external potential is applied. This potential may be induced, e.g., by a homogeneous electric field $\mathbf{E}$ and if the weathercock is dielectric with an additional permanent electric charge placed somewhere else than on the axis $\mathbf{e}_z$ the external potential (26) is achieved.

## 1 Introduction

In the last couple of years there has been enormous progress in the field of levitated optomechanics [1, 2] that now allows for mesoscopic massive particles to enter the center-of-mass quantum regime down to several quanta and even the motional ground state [3–7]. Simultaneously, the improved manipulation of orientational degrees of freedom [8–15] promises to soon reach the angular momentum ground states of nano-scale rotors [16–22]. For the latter, the correct description of decohering system-bath interactions are most important for ambitious proposals to test quantum-classical boundaries [23, 24] but are also relevant for nano-technological applications [2, 25–28]. While models describing decoherence, diffusion, and thermalization of the center-of-mass degrees of freedom with an environment have been well established [29, 30], open quantum systems involving orientational degrees of freedom have only recently been tackled successfully [31]. Still, up to this date, studies on diffusion of orientational degrees of freedom, even for the simplest case of a planar rotor, are often restricted to linear, i.e. Cartesian, asymptotic solutions around equilibrium positions, which entails a large numerical overhead and further approximations [32–34]. We will address this problem of orientational thermalization in one dimension in its most general manifestation, as depicted in Fig. 1, including an external potential induced by, e.g., an electric field. Apart from the aforementioned field of levitated optomechanics, our results will also be useful to describe the environmental influence on rotational based nanomachines and heat engines [35–39].

The model discussed in this article is based on a recently found thermalization master equation for asymmetric rotors [31], derived from the Caldeira-Leggett master equation for the constituent point particles [29, 40], and will be implemented in a general kinematic model. To introduce this model as instructively as possible, we present the equations of motion in terms of the well-known Wigner function in phase space [41–43]. This quasi probability distribution is mostly used for linear motion and harmonic oscillators in Cartesian phase space

and enjoys great popularity due to the close connection to its classical pendant while depicting quantum signatures elegantly. For example, the Wigner functions associated to spatial superposition states or Fock states exhibit areas with negative sign that would be classically forbidden [44]. The similarities to the classical phase space distribution are also reflected in the kinematic equations, which resemble the classical ones for linear dynamics (i.e., at most harmonic potentials) and otherwise add higher-order quantum corrections that allow for an easy identification of the classical limit [45].

This simple intuitive picture does not translate to the periodic phase space of a quantum rotor when using the proper periodic Wigner function first derived by Mukunda [46,47]: Already in the most elementary case of planar rotation with a single angular degree of freedom, the periodicity of the angle leads to complications in the kinematic description [48,49]. As a remedy, one can decompose the proper Wigner function into auxiliary functions [48] which on their own no longer fulfill the desired properties of a phase space function; however, their time evolution is straightforward and resembles the classical intuition just as in the case of the Cartesian Wigner function. Hence these auxiliary functions are most suited to discuss the thermalization process as a result of the combined diffusion and friction in presence of an external potential.

In this article we expand the results of Ref. [31] for planar rotations to the more general case including external potentials. We further express the thermalization process in terms of the auxiliary Wigner functions to also describe rotors that are not symmetric under inversion. The formulas presented throughout the article are meant to provide a complete mathematical toolbox to implement decoherence, diffusion, and friction for the general planar rotor.

The remainder of this article is structured as follows: In Sec. 2 we will re-visit the one-dimensional and periodic Wigner phase space distribution together with the auxiliary Wigner functions as introduced by Bizarro [48]. This will be the basis of the thermalization kinematics presented in Sec. 3. We will show analytical results for the model in Sec. 4 and numerical simulations for exemplary scenarios in Sec. 5, followed by a concluding Sec. 6.

# 2  Wigner-Weyl formalism for periodic phase space

This section, together with a majority of Sec. 3, serves as an introduction to the theoretical framework and is based on previous work. Readers familiar with the topic of periodic phase space may continue with Eq. (18).

## 2.1  Cartesian case

Since quantum mechanics is inherently a probabilistic theory, it is natural to compare it to classical mechanics on the level of probability distributions in phase space [43]. The most prominent and widely used approach was introduced by Weyl [41] and Wigner [42]. In the case of one-dimensional motion, for example, the Wigner-Weyl formalism maps the quantum mechanical state operator $\rho$ to a quasi phase-space distribution

$$
\begin{aligned}
W(x,p) &= \frac{1}{\pi\hbar}\int \mathrm{d}s\, e^{-2ips/\hbar}\langle x+s|\rho|x-s\rangle \\
&= \frac{1}{\pi\hbar}\int \mathrm{d}q\, e^{2ixq/\hbar}\langle p+q|\rho|p-q\rangle\,,
\end{aligned} \tag{1}
$$

at position $x$ and momentum $p$ and the eigenstates $|x\rangle$ and $|p\rangle$ of the respective operators. Even though the Wigner function can become negative, a feature widely recognized to show the

quantum nature of a state [44], it fulfills the desired marginalization rules and normalization of a phase space distribution,

$$\int \mathrm{d}x\, W(x,p) = \langle p|\rho|p\rangle\,,$$

$$\int \mathrm{d}p\, W(x,p) = \langle x|\rho|x\rangle\,,$$

$$\int \mathrm{d}x\, \mathrm{d}p\, W(x,p) = 1\,. \tag{2}$$

A given time evolution of the quantum state, $\partial_t \rho = \mathcal{L}\rho$, described by a physical superoperator $\mathcal{L}$, can be converted to a partial differential equation for the Wigner function via

$$\partial_t W(x,p) = \frac{1}{\pi\hbar}\int \mathrm{d}s\, e^{-2ips/\hbar}\langle x+s|\mathcal{L}\rho|x-s\rangle\,. \tag{3}$$

For a closed system evolving unitarily under an external potential, one arrives at the quantum Liouville equation [50].

## 2.2 Periodic boundary conditions

Let us now consider a planar rotor in phase space, as described by a periodic angular coordinate $\alpha$ and its associated canonical angular momentum $p_\alpha$. When quantizing these canonical variables, the easiest way to enforce the periodic boundary conditions, $\alpha \in [-\pi,\pi)$, is to introduce and always use the orientation operator $e^{i\hat{\alpha}}$ instead of $\hat{\alpha}$ itself, thus avoiding issues regarding the fundamental uncertainty principle between $\hat{\alpha}$ and $\hat{m} = \hat{p}_\alpha/\hbar = -i\partial_\alpha$ [48,51].

The finite configuration space can be represented by the basis of improper eigenvectors $|\alpha\rangle$ of the orientation operator, $e^{i\hat{\alpha}}|\alpha\rangle = e^{i\alpha}|\alpha\rangle$ or, alternatively, by the proper orthonormal basis of *discrete* angular momentum eigenstates $|m\rangle$ with $m \in \mathbb{Z}$. The two bases are related by a periodic Fourier transformation, through $\langle\alpha|m\rangle = e^{i\alpha m}/\sqrt{2\pi}$. Note that the $|\alpha\rangle$ are strictly only defined for $\alpha \in [-\pi,\pi)$, and we shall implicitly assume throughout this article that $\alpha \notin [-\pi,\pi)$ be replaced by $\mathrm{mod}(\alpha+\pi,2\pi)-\pi$ for periodic continuity.

The quantum state of a planar rotor can be represented in phase space by the periodic Wigner function

$$W(\alpha,m) = \frac{1}{\pi}\int_{-\pi/2}^{\pi/2}\mathrm{d}\alpha' e^{-2im\alpha'}\langle\alpha+\alpha'|\rho|\alpha-\alpha'\rangle\,, \tag{4}$$

which was already proposed in the pioneering work of Mukunda [47] and relates to the Cartesian case (1) in a seemingly straightforward manner. Indeed, the periodic Wigner function (4) yields the correct angle and momentum marginals

$$\int_{-\pi}^{\pi}\mathrm{d}\alpha\, W(\alpha,m) = \langle m|\rho|m\rangle\,,$$

$$\sum_m W(\alpha,m) = \langle\alpha|\rho|\alpha\rangle\,,$$

$$\sum_m \int_{-\pi}^{\pi}\mathrm{d}\alpha\, W(\alpha,m) = 1\,, \tag{5}$$

analogous to the Cartesian case (2). However, if we Fourier-transform the density matrix on

the right hand side of (4) to momentum representation, we find

$$W(\alpha, m) = \frac{1}{2\pi} \sum_{m_1, m_2} \text{sinc}\left[\left(m - \frac{m_1 + m_2}{2}\right)\pi\right] \times e^{i(m_1 - m_2)\alpha}\langle m_1|\rho|m_2\rangle \tag{6}$$

$$\neq \frac{1}{2\pi} \sum_{m'} e^{2i\alpha m'}\langle m + m_1|\rho|m - m'\rangle, \tag{7}$$

which does *not* resemble the discrete version of the momentum integral in the Cartesian case shown in the second line of (1). As a significant consequence, the quantum time evolution of a free planar rotor no longer matches the classical evolution in phase space: a shearing transformation periodically wrapped to $\alpha \in [-\pi, \pi)$. As we will show in Sec. 3, this mismatch can be alleviated by expanding the Wigner function into auxiliary Wigner functions [48] associated to integer and half-integer values of $(m_1 + m_2)/2$ in the double sum of (6),

$$W(\alpha, m) = W_m(\alpha) + \sum_{m'} \text{sinc}\left[\left(m - m' - \frac{1}{2}\right)\pi\right] W_{m'+1/2}(\alpha). \tag{8}$$

The integer and the half-integer auxiliary terms are given, respectively, by $\pi$-periodic and $2\pi$-periodic phase-space functions, defined as [48]

$$W_{m+\mu/2}(\alpha) = \frac{1}{2\pi} \int_{-\pi}^{\pi} d\alpha' e^{-2i(m+\mu/2)\alpha'}\langle \alpha + \alpha'|\rho|\alpha - \alpha'\rangle, \tag{9}$$

for $\mu \in \{0, 1\}$. Crucially, the marginals and normalization of these auxiliary functions (9) now read

$$\int_{-\pi}^{\pi} d\alpha\, W_m(\alpha) = \langle m|\rho|m\rangle,$$

$$\sum_m [W_m(\alpha) + W_{m+1/2}(\alpha)] = \langle \alpha|\rho|\alpha\rangle,$$

$$\sum_m \int_{-\pi}^{\pi} d\alpha\, W_m(\alpha) = 1, \tag{10}$$

rendering them unsuitable as a quasi probability distribution, but they will allow us to calculate the time evolution of the quantum rotor state in phase space in a concise manner. For convenience, we shall abbreviate the half-integer index $\nu = m + \mu/2$ from this point onward.

## 3 Kinematic equations

We want to describe the open quantum dynamics of a planar rotor with moment of inertia $I$ under the influence of an external potential and environmental friction and diffusion leading to thermalization. In the Schrödinger picture, the time evolution of the quantum planar rotor state is

$$\partial_t \rho = \frac{1}{i\hbar}[\hat{T} + \hat{V}, \rho] + \mathcal{L}\rho, \tag{11}$$

where $\hat{T} = \hat{p}_\alpha^2/2I$ is the kinetic energy, $\hat{V} = V(\hat{\alpha})$ a periodic potential energy, and $\mathcal{L}\rho$ a Lindblad term describing thermalization, all specified further below. Switching to the phase space representation and expanding the Wigner function according to (9), we identify the respective contributions to the time evolution of each auxiliary Wigner function as

$$\partial_t W_\nu(\alpha) = (\partial_t^T + \partial_t^V + \partial_t^{\mathcal{L}})W_\nu(\alpha). \tag{12}$$

In the following subsections, we will discuss the terms individually.

## 3.1 Unitary time evolution

The term $\partial_t^T W_\nu(\alpha)$ stems from the kinetic energy part $\hat{T} = \hat{p}_\alpha^2/2I$ of the free Hamiltonian and reads [48]

$$\partial_t^T W_\nu(\alpha) = -\frac{\nu\hbar}{I}\partial_\alpha W_\nu(\alpha). \tag{13}$$

This now matches the free time evolution generator for a classical planar rotor or particle on a line in phase space. It yields the (periodically wrapped) shearing transformation, $W_\nu(\alpha, t) = W_\nu(\alpha - \nu\hbar t/I, 0)$, for each auxiliary Wigner function (9), facilitating an analytical treatment of quantum rotations in phase space. The actual Wigner function (8) is a sum of different auxiliary terms and its free time evolution is therefore *not* a simple shearing transformation.

Physically, the more complex behavior of the free quantum time evolution in orientational phase space, as opposed to a mere shearing in the Cartesian (or classical) case, is closely related to interference effects in periodic phase space, which also lead to quantum state revivals at multiples of the revival time $t_r = 4\pi I/\hbar$. We illustrate this by exemplary snapshots of the Wigner function in Fig. 2. The combination of Eqs. (13) and (8) lead to the emergence of negativities in the phase space distribution (indicating the quantum nature of the state), which result in interference patterns in the marginals and (partial) revivals of the initial quantum state.[1] The (partial) revivals of massive quantum rotors are a promising new avenue for applications and tests of mesoscopic quantum phenomena [24, 28], and for this their correct description in presence of time-dependent potentials and dissipative channels is imperative.

Now let the rotor be subject to a time-independent external potential. Adhering to the rotational symmetry, the potential is a generic $2\pi$-periodic function that can be Fourier-expanded as

$$\hat{V} = \sum_{k=1}^{\infty}[a_k\cos k\hat{\alpha} + b_k\sin k\hat{\alpha}]. \tag{14}$$

Note that even though we do not consider $a_k(t)$, $b_k(t)$ here, because they would prohibit thermalization, the model could just as well be implemented in more elaborate interference schemes based on time dependent potentials [28]. The associated contribution to the time evolution of the auxiliary Wigner functions in phase space reads

$$\partial_t^V W_\nu(\alpha) = \frac{1}{\hbar}\sum_{k=1}^{\infty}[a_k\sin k\alpha + b_k\cos k\alpha] \times \left[W_{\nu-k/2}(\alpha) - W_{\nu+k/2}(\alpha)\right], \tag{15}$$

and a similar form is obtained for the contribution to the time evolution of the proper Wigner function (4) in this case.

## 3.2 Thermalization Lindbladian

With the unitary time evolution in phase space at hand, we now consider the phase space representation of the Lindbladian that generates thermalization. For a planar rotor, the generator can be written as [31]

$$\mathcal{L}\rho = \frac{2D}{\hbar^2}[\mathbf{e}_r(\hat{\alpha})\cdot\rho\mathbf{e}_r(\hat{\alpha}) - \rho] + \frac{i\Gamma}{2\hbar}[\mathbf{e}_\varphi(\hat{\alpha})\hat{p}_\alpha\rho\mathbf{e}_r(\hat{\alpha}) - \mathbf{e}_r(\hat{\alpha})\cdot\rho\hat{p}_\alpha\mathbf{e}_\varphi(\hat{\alpha})]$$
$$+ \frac{D}{8k_B^2T^2I^2}\left[\mathbf{e}_\varphi(\hat{\alpha})\hat{p}_\alpha\cdot\rho\hat{p}_\alpha\mathbf{e}_\varphi(\hat{\alpha}) - \frac{1}{2}\{\hat{p}_\alpha^2, \rho\}\right], \tag{16}$$

---

[1]Note that one would obtain a simple shearing transformation as well as revivals of the proper Wigner function, albeit with the wrong periodicity, if one wrongfully imposed the momentum representation (7).

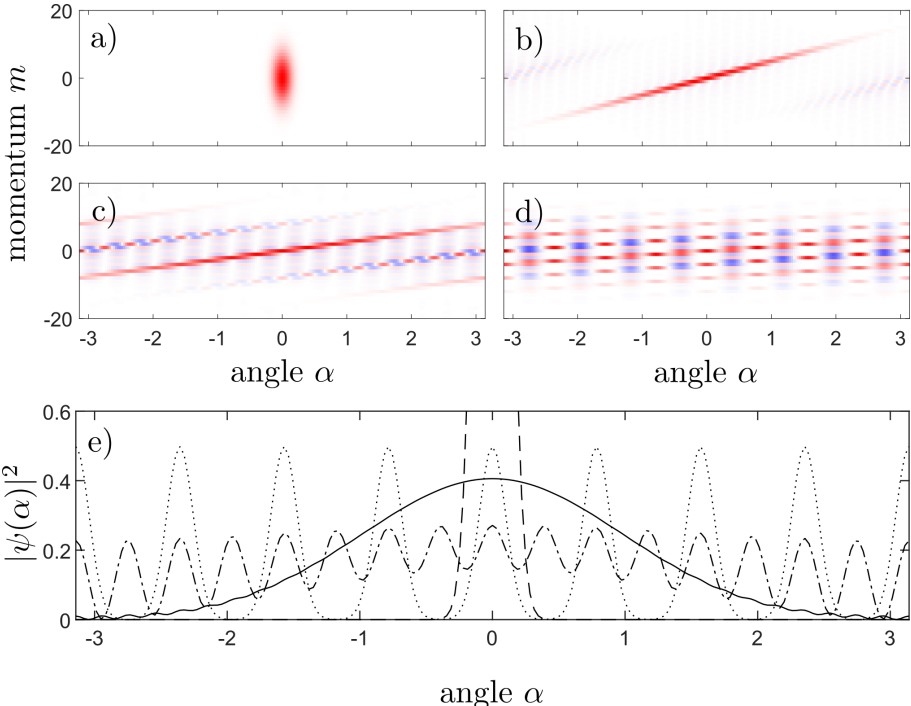

Figure 2: (a)-(d) Exemplary snapshots of the Wigner function and (e) their marginal angular distributions for a freely rotating planar rotor. The initial state in (a) and its marginal (dashed curve) correspond to an approximately Gaussian wave packet of width $\sigma = 0.1$ centered at $\alpha = 0$, as defined in (27). The later snapshots (b), (c), and (d), and their marginals represented by the solid, dash-dotted, and dotted curves in (e), are evaluated at the fractions $t = t_r/64$, $t_r/32$, and $t_r/16$ of the revival time $t_r = 4\pi I/\hbar$, respectively. For better visibility, each color scale in (a)-(d) is normalized to the maximum (red) and the negative minimum (blue) of the Wigner function. The visible negativities in (c) coincide with the appearance of interference fringes in the marginal (dash-dotted).

with the Boltzmann constant $k_B$, the bath temperature $T$, the moment of inertia $I$ of the rotor, the friction rate $\Gamma$, and the diffusion coefficient $D = \Gamma k_B T I$. We also introduce the orientation vector $\mathbf{e}_r(\alpha) = (\cos\alpha, \sin\alpha)^{\mathrm{T}}$ in the rotation plane and its associated velocity direction vector $\mathbf{e}_\varphi(\alpha) = \partial_\alpha \mathbf{e}_r(\alpha)$.[2] Thus, by construction, the angle operator $\hat{\alpha}$ only ever appears inside well-defined periodic functions.

The first line in Eq. (16) describes momentum diffusion, the second line friction, and the third line angular diffusion. The latter, while needed to ensure the complete positivity of the time evolution, is strongly suppressed by $\hbar^2$ and can therefore be neglected in the classical limit where the thermal energy greatly exceeds kinetic energy quanta, $I k_B T/\hbar^2 \gg 1$. The first and the second moment of angular momentum evolve under the dissipator as

$$\left\langle \mathcal{L}^\dagger \hat{p}_\alpha \right\rangle = -\Gamma \left\langle \hat{p}_\alpha \right\rangle, \qquad \left\langle \mathcal{L}^\dagger \hat{p}_\alpha^2 \right\rangle = 2D - 2\Gamma \left\langle \hat{p}_\alpha^2 \right\rangle, \tag{17}$$

which describes a constant increase of the kinetic energy due to diffusion and a state-dependent linear friction. Together, they lead to approximate thermalization, i.e., equilibration into a state very close to the Gibbs state $\rho_G \sim \exp[-\hat{T}/k_B T]$ if no potential is present [31].

---

[2]The reader my have noticed that the generator (16) lacks the Hamiltonian-like term known from the Caldeira-Leggett dissipator in Cartesian phase space [40]. A similar term does show up in the derivation of (16), see Ref. [31], but it ultimately drops out due to $\mathbf{e}_\varphi(\hat{\alpha}) \cdot \mathbf{e}_r(\hat{\alpha}) = 0$.

Table 1: Moment of inertia, collisional diffusion rate and final thermal occupation number for typical proposed and realized nano-rotors.

| | $I\,[\mathrm{kg\,m^2}]$ | $2D_g/\hbar^2\,[\mathrm{s^{-1}}]$ | $m_{\mathrm{th}}$ |
|---|---|---|---|
| Stickler et al. [24] silicon rod | $4.84 \cdot 10^{-37}$ | 24.8 | $6.0 \cdot 10^6$ |
| Stickler et al. [24] carbon nanotubes | $6.57 \cdot 10^{-38}$ | 7.0 | $2.2 \cdot 10^6$ |
| Pontin et al. [21] silica ellipsoid | $6.78 \cdot 10^{-32}$ | $5.9 \cdot 10^3$ | $2.2 \cdot 10^8$ |
| Bang et al. [18] nanodumbbells | $3.37 \cdot 10^{-32}$ | $3.9 \cdot 10^3$ | $1.6 \cdot 10^8$ |

In phase space, the dissipator (16) acts on the auxiliary Wigner functions like

$$\partial_t^{\mathcal{L}} W_\nu(\alpha) = \frac{D}{\hbar^2}\left[W_{\nu+1}(\alpha) + W_{\nu-1}(\alpha) - 2W_\nu(\alpha)\right] + \frac{\Gamma}{2}\left[(\nu+1)W_{\nu+1}(\alpha) - (\nu-1)W_{\nu-1}(\alpha)\right]$$
$$+ \frac{\hbar^2\Gamma}{16 k_B T I}\left[\left(\frac{\partial_\alpha^2}{4} + (\nu+1)^2\right)W_{\nu+1}(\alpha) + \left(\frac{\partial_\alpha^2}{4} + (\nu-1)^2\right)W_{\nu-1}(\alpha) - 2\left(\frac{\partial_\alpha^2}{4} + \nu^2\right)W_\nu(\alpha)\right], \quad (18)$$

kicking off the original results of this work. Here, the two terms in the first line describe second- and first-order discrete momentum derivatives representing quantized momentum diffusion and friction, respectively. The second line represents the minor correction due to angular diffusion, suppressed by $\hbar^2/k_B T I$ classical high-temperature limit, but needed for complete positivity.

In Tab. 1 we list the relevant experimental parameters for proposals in the quantum regime [24] and selected realizations of trapped nano-rotors [18, 21], calculating the most relevant gas diffusion constant [24, 52] $D_g \simeq \sqrt{\pi/2}\hbar^2 p_g d\ell(1 + d/\ell)/\sqrt{m_g k_B T}$ with the rotor length $\ell$, diameter $d$, gas pressure $p_g$ and gas particle mass $m_g$ (assuming nitrogen in each case at room temperature $T = 300\,\mathrm{K}$ and $p_g = 5 \cdot 10^{-9}\,\mathrm{mbar}$). In all cases the thermal occupation number $m_{\mathrm{th}} = p_{\mathrm{th}}/\hbar = \sqrt{2k_B T I/\hbar^2}$ is deep in the high temperature regime. To thermalize on few-digits $m_{\mathrm{th}}$ one would require a moment of inertia as small as exhibited by molecules and/or sub-Kelvin temperatures. In this case other thermalization sources like background radiation would be required for the Caldeira-Leggett approximation to remain valid.

## 4 Analytical results

In general, the time evolution of the quantum rotor state described by the von Neumann equation (11) or its counterpart in the Wigner representation can only be solved numerically. However, thanks to the representation in terms of the auxiliary Wigner functions in (9), we can obtain approximate analytical results for the classical high-temperature limit.

First, we show that the Gibbs state $\rho_G \propto e^{-\hat{H}/k_B T}$ with $\hat{H} = \hat{T} + \hat{V}$ is the steady state of the time evolution up to leading order in $\epsilon_1 = \hbar^2/k_B T I$ and $\epsilon_2 = \hbar^2 \max_\alpha\{V''(\alpha)\}/k_B^2 T^2 I$. That is, the dissipator (16) thermalizes the rotor state in the presence of a potential at sufficiently high temperatures such that $\epsilon_{1,2} \ll 1$. To this end, we follow the same path as we did for the free rotor in Ref. [31] and rewrite the dissipator (16) such that its Lindblad form becomes

explicit. Applied to the Gibbs state, and with $D = \Gamma k_B T I$ inserted, we get

$$
\begin{aligned}
\mathcal{L}\rho_G &= \frac{2D}{\hbar^2}\left[\hat{A}\cdot\rho_G\hat{A}^\dagger - \frac{1}{2}\left\{\hat{A}^\dagger\cdot\hat{A},\rho_G\right\}\right] \\
&= \frac{2\Gamma}{\epsilon_1}\left[\hat{A}\cdot F(\hat{A}^\dagger) - \frac{1}{2}\hat{A}^\dagger\cdot\hat{A} - \frac{1}{2}F(\hat{A}^\dagger\cdot\hat{A})\right]\rho_G,
\end{aligned}
\tag{19}
$$

with the Lindblad operator

$$
\hat{A} = \mathbf{e}_r(\hat{\alpha}) + \frac{i\hbar}{4k_B T I}\mathbf{e}_\varphi(\hat{\alpha})\hat{p}_\alpha = \mathbf{e}_r(\hat{\alpha}) + \frac{i\epsilon_1}{4\hbar}\mathbf{e}_\varphi(\hat{\alpha})\hat{p}_\alpha.
\tag{20}
$$

The second line in Eq. (19) follows by multiplying from the right with $e^{\hat{H}/k_B T}e^{-\hat{H}/k_B T}$ and expanding $e^{-\hat{H}/k_B T}\hat{A}^{(\dagger)}e^{\hat{H}/k_B T}$ via

$$
F(\hat{B}) = \sum_{k=0}^{\infty}\frac{(-k_B T)^{-k}}{k!}[\hat{H},\hat{B}]_k,
\tag{21}
$$

where $[\hat{H},\hat{B}]_k = [\hat{H},[\hat{H},\ldots,[\hat{H},\hat{B}]\ldots]]$ denotes the $k$-fold commutator.[3] The unitary part of the time evolution is generated by $\hat{H}$ and thus leaves the Gibbs state invariant by construction. Hence we are left with showing that $\mathcal{L}\rho_G$ vanishes when $\epsilon_{1,2}\to 0$.

At first sight, (19) seems to diverge in the high-temperature limit. However, evaluating the kinetic energy terms and the leading contributions of $\hat{V}$ in the expression (21), we find that all critical terms in (19) proportional to $\epsilon_1^{-1}$ and $\epsilon_1^0$ cancel, and the remaining leading-order corrections are of the orders $\epsilon_1$ and $\epsilon_2$, see the appendix for details. In order to understand the physical meaning of $\epsilon_2$, it is instructive to consider a strong trapping potential that could align the rotor, say, at around $\alpha = 0$. A second-order expansion of the potential yields the characteristic trapping frequency $\Omega = \sqrt{V''(0)/I}$, and so $\epsilon_2 \sim (\hbar\Omega/k_B T)^2$. Hence we see that $\epsilon_{1,2}\to 0$ corresponds to the limit in which thermal excitations reach far beyond the deep quantum regime.

We can also verify that the continuous Fokker-Planck equation is re-obtained in the classical limit. To this end, we apply the continuum limit to the discrete angular momentum numbers, $p_\alpha = \hbar m$ with $\hbar \to 0$, which effectively removes all half-integer auxiliary functions terms the Wigner function expansion in the second line of (8),

$$
\lim_{\hbar\to 0}\frac{1}{2}\int_{-\infty}^{\infty}dp'_\alpha\frac{1}{\hbar}\mathrm{sinc}\left[\pi\left(\frac{p_\alpha - p'_\alpha}{\hbar} - \frac{1}{2}\right)\right]W_{p'_\alpha/\hbar + 1/2}(\alpha) = \frac{1}{2}W_{p_\alpha/\hbar}(\alpha).
\tag{22}
$$

Here we used the sinc-function representation of the delta distribution. Thus, with the half-integer terms absent, we can employ the same techniques as in Ref. [52] to obtain the approximate Fokker-Planck equation

$$
\begin{aligned}
\partial_t W(p_\alpha,\alpha,t) \simeq &-\frac{p_\alpha}{I}\partial_\alpha W(p_\alpha,\alpha,t) + V'(\alpha)\partial_{p_\alpha}W(p_\alpha,\alpha,t) \\
&+ \Gamma\partial_\alpha[p_\alpha W(p_\alpha,\alpha,t)] + D\partial_{p_\alpha}^2 W(p_\alpha,\alpha,t).
\end{aligned}
\tag{23}
$$

It no longer contains the angular diffusion term (of order $\epsilon_1$), and it replaces discrete momentum differences from Eq. (18) by first- and second-order derivatives. Naturally, one can also

---

[3]Note that this Baker-Campbell-Hausdorff decomposition is not rigorous: the sum does not converge due to unbound operators. We can simply truncate the Hilbert space at a suitable $p_{\alpha,\mathrm{trunc.}} \gg \max\{\sqrt{2k_B T I},\sqrt{2V(\alpha)I}\}$ which renders Eq. (21) correct and only introduces a minor error that is exponentially suppressed with $\exp(-p_{\alpha,\mathrm{trunc.}}^2/2k_B T)$ when eventually applying the Gibbs state.

show that the Wigner representation of the continuous version of the Gibbs state $\rho_G$ is a steady state of the Fokker-Planck equation (23).

Finally, we can give a simple analytic expression for the time-evolved phase-space state of a free quantum rotor ($V = 0$) when only frictionless diffusion is present ($\Gamma \to 0$, but $D > 0$), as asymptotically realized by an infinite-temperature bath. Given any initial state specified by the auxiliary functions $W_\nu(\alpha, 0)$, the auxiliary functions at later times are obtained by a combination of shearing and convolution,

$$W_\nu(\alpha, t) = \sum_{\ell \in \mathbb{Z}} \int_{-\pi}^{\pi} d\alpha' \, W_{\nu-\ell}(\alpha - \alpha' - \hbar t \nu/I, 0) K_\ell(\alpha', t), \tag{24}$$

with the kernels

$$K_\ell(\alpha', t) = \frac{1}{2\pi} e^{-2Dt/\hbar} \times \sum_{k \in \mathbb{Z}} e^{ik(\alpha' - \hbar t \ell/2I)} I_\ell \left[ \frac{2Dt}{\hbar^2} \text{sinc}\left( \frac{\hbar k t}{2I} \right) \right]. \tag{25}$$

Here, $I_\ell(\cdot)$ is a modified Bessel function of the first kind. The solution preserves the norm of each auxiliary function and the Wigner function because of $\sum_\ell \int d\alpha' K_\ell(\alpha', t) = 1$, and it generalizes our earlier results for inversion-symmetric rotors reported in Refs. [23, 52].

# 5 Numerical simulations

Apart from the specific analytical results presented in the previous section, the time evolution (12) for a general quantum state of the planar rotor has to be calculated numerically. In this section we will demonstrate the thermalization process for an exemplary potential and initial state and show deviations between the here developed quantum mechanical model and its classical counterpart. All our numerical results can be expressed in terms of the dimensionless parameters $\tilde{t} = t \sqrt{V_0/I}$, $\tilde{T} = k_B T/V_0$, and $\tilde{\hbar} = \hbar/\sqrt{V_0 I}$, where $V_0$ denotes the characteristic strength of the external potential and $\tilde{\hbar} \to 0$ effectively marks the classical limit.

First proof-of-principle experiments are exclusively focused on inversion symmetric particles trapped in potentials with two identical global minima and $I/V_0 \to 0$. Instead, to demonstrate the full capacity of Eq. (18), we will now show the time evolution of much harder to prepare (superposition) states in slightly more elaborate potentials and $\tilde{T}, \tilde{\hbar}$ close to unity. Let us assume a $2\pi$-periodic potential of the form

$$V(\alpha) = V_0(\cos \alpha - \cos 2\alpha), \tag{26}$$

which has a local minimum at $\alpha = 0$ and a global minimum at $\alpha = \pm\pi$. Note that the potential is not $\pi$-periodic, and therefore the rotor is not inversion-symmetric. This means that both the integer and the half-integer auxiliary Wigner functions must be taken into account in the expansion (8).

Let us further assume a pure initial state that resembles the periodic equivalent of a Gaussian wave packet,

$$\langle \alpha | \psi \rangle = \frac{1}{N} \exp\left[ -\frac{1}{\sigma^2} \sin^2\left( \frac{\alpha - \alpha_0}{2} \right) \right], \tag{27}$$

with the normalization factor $N = \sqrt{2\pi I_0(1/\sigma^2) e^{-1/\sigma^2}}$. We will work with small variances $\sigma^2 \ll 1$ so that the state is practically a Gaussian function in $\alpha$, as one would obtain for a rotor that is deeply trapped in an approximately harmonic potential. The specific form of the wave packet allows us to give explicit expressions for the corresponding initial Wigner function (4),

$$W(\alpha, m) = \sum_k \text{sinc}\left[ \left( m + \frac{k}{2} \right) \pi \right] \frac{I_k \left[ \cos(\alpha - \alpha_0)/\sigma^2 \right]}{2\pi I_0[1/\sigma^2]}, \tag{28}$$

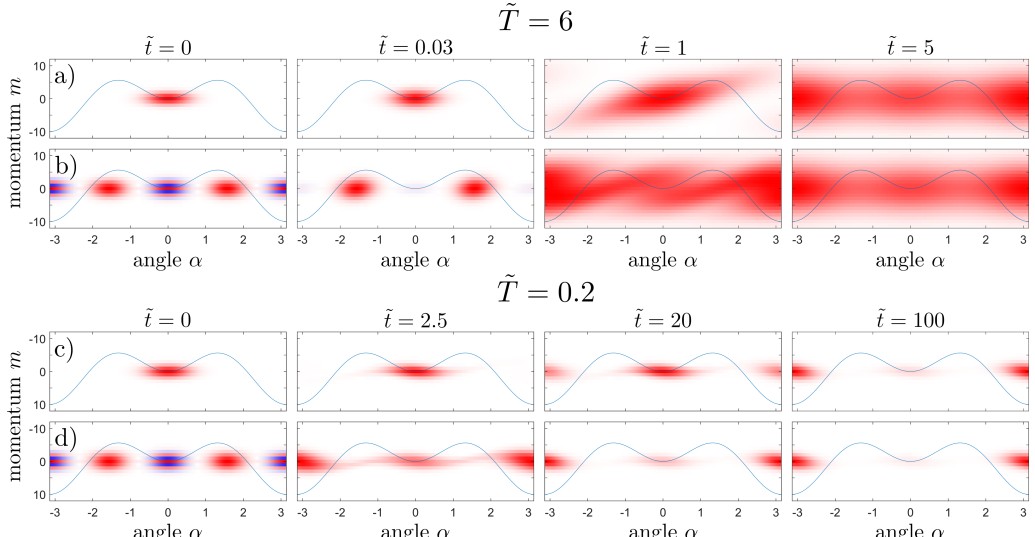

Figure 3: Density plots of Wigner function snapshots for the time evolution of two different initial rotor states subjected to the potential (26) and to thermalization. We juxtapose the high-temperature regime at $\tilde{T} = 6$, shown in (a) and (b), to the low-temperature regime at $\tilde{T} = 0.2$ in (c) and (d). In (a) and (c), the initial rotor state is a Gaussian-like wave packet (27) with $\sigma = 0.4$ and $\alpha_0 = 0$, whereas in (b) and (d), we consider an equal superposition of two such wavepackets centered around $\alpha_0 = \pm\pi/2$ with $\sigma = 0.3$. In each diagram, the color scale is normalized to the maximum (red) and the minimum (blue) value of the Wigner function. The blue line shows the shape of the potential (26) for reference. The potential strength and the moment of inertia are chosen such that $\tilde{\hbar} = 0.5$ and $\Gamma = \sqrt{V_0/I}$.

and identify the auxiliary Wigner functions as

$$W_\nu(\alpha) = \frac{I_{2\nu}[\cos(\alpha - \alpha_0)/\sigma^2]}{2\pi I_0[1/\sigma^2]}. \tag{29}$$

In order to compute the time-evolved state, we now simply propagate the $W_\nu(\alpha)$ according to (12).

In Fig. 3, we illustrate the thermalization process for two different temperatures in the classical regime and in the deep quantum regime, and for two different initial states: a quasi-Gaussian wave packet at the local minimum ($\alpha_0 = 0$), and a superposition of such wave packets at $\alpha_0 = \pm\pi/2$, i.e., close to the potential maxima. As a first observation, the decoherence induced by the dissipator (18), which destroys the oscillating phase-space negativities of the superposition state in (b) and (d), takes place on much shorter time scales than the actual dissipation leading to the equilibrium state. Note that, because we have chosen an appreciable friction rate $\Gamma = \sqrt{V_0/I}$ here, the Wigner function barely shears under the unitary evolution before it is decohered. In the opposite regime of small $\Gamma$, one could observe the shearing of the distribution, and possibly also interference fringes as in Fig. 2.

In the high-temperature case ($\tilde{T} = 6$) shown in (a) and (b), the final phase space distribution at $\tilde{t} \gg 1$ ($t \gg \sqrt{I/V_0}$) will be indistinguishable from the Gibbs state $\rho_G$. Even though this temperature is large enough for the state to be unbounded by the potential, leading to a smeared out Wigner function over the whole orientation space, the modulation due to the potential's shape is still visible in the right-most panels at $\tilde{t} = 5$. At even higher temperatures, the influence of the potential becomes negligible and the asymptotic steady state would converge

to the one of a free planar rotor [31],

$$\rho_{\text{eq}} \simeq \sum_m \frac{1}{8^{\tilde{T}/\tilde{\hbar}^2}} \begin{pmatrix} 4\tilde{T}/\tilde{\hbar}^2 \\ 2\tilde{T}/\tilde{\hbar}^2 + m \end{pmatrix} |m\rangle\langle m|, \tag{30}$$

where the $\binom{n}{k}$ denote binomial coefficients and $\tilde{T}/\tilde{\hbar}^2 = k_B T I/\hbar^2$.

For the low temperature $\tilde{T} = 0.2$ in (c) and (d), the equilibrium state will be eventually localized in the global minimum of the potential. However, if the initial state is trapped in the local minimum as in (c), it may take a longer time to reach the equilibrium, because thermal excitations are suppressed and the state thus needs to tunnel through the potential barriers to reach the global minimum. The slow-down of the equilibration can be seen by observing the panels in (c) and (d) at intermediate times. In a classical rotor model without tunneling, the initial state in (c) would be meta-stable and thermalization inhibited.

The asymptotic steady state always deviates from the Gibbs state $\rho_{\text{G}}$, especially in the low-temperature regime where only few angular momenta $m$ are populated. We illustrate this in Fig. 4 for the potential (26) by plotting as a function of $\tilde{T}$ the trace distance between the actual equilibrium state $\rho_{\text{eq}}$ that we obtain numerically and the Gibbs state $\rho_{\text{G}}$,

$$d_1(\rho_{\text{eq}}, \rho_{\text{G}}) = \frac{1}{2}\text{tr}\left[\sqrt{(\rho_{\text{eq}} - \rho_{\text{G}})^\dagger(\rho_{\text{eq}} - \rho_{\text{G}})}\right] \in [0, 1]. \tag{31}$$

The deviations from the Gibbs state decrease once like $\tilde{T}^{-2}$ at medium temperatures where the equilibrium state is still affected by the potential and once like $\tilde{T}^{-1}$ for large temperatures where the potential becomes negligible and the rotor behaves quasi free. This is in accordance with our analysis in Sec. 4, as $d_1(\rho_{\text{eq}}, \rho_{\text{G}}) = \mathcal{O}(\epsilon_1, \epsilon_2)$ and $\epsilon_2 \ll \epsilon_1$ for $\tilde{T} \to \infty$. At small temperatures, however, the deviation from the Gibbs state can be significant; indeed, the trace distance almost reaches its maximum for the exemplary case of $\tilde{\hbar} = 1$, far from the classical limit $\tilde{\hbar} \ll 1$ ($V_0 \gg \hbar^2/I$).

Finally, we would like to comment on the role of the angular diffusion term proportional to $\hbar^2$ in the dissipator (18). This term has no classical equivalent and comes from an ad-hoc correction of the Caldeira-Leggett master equation to ensure complete positivity. For a free rotor, it would make a relevant contribution to the dynamics only at very low temperatures close to the quantum ground state—a regime in which the approximations underlying the Caldeira-Leggett master equation would break down anyway. In the presence of a potential, however, the interplay between the potential's contribution to the evolution and the angular diffusion can affect the asymptotic behavior also at higher temperatures. Specifically, the precise equilibrium state may then depend on the friction rate $\Gamma$, whereas classically and for a free rotor, the rate determines merely how fast the rotor relaxes. An experimental investigation of the possible $\Gamma$-dependence at equilibrium could shed light on the angular diffusion term and clarify whether it is of physical origin.

# 6 Conclusion

We presented the general model of one-dimensional thermalization in presence of an external potential for periodic degrees of freedom. We demonstrated analytical results to verify important key features of the thermalization process: as more and more quanta of angular momentum are occupied with growing temperature, the Gibbs state becomes a good approximation for the equilibrium state of the quantum rotor, and this coincides with the classical limit in which the phase space representation of the thermalization master equation is well described by a continuous Fokker-Planck equation. These results are supported by our numerical studies for different scenarios, demonstrating not only decoherence and thermalization,

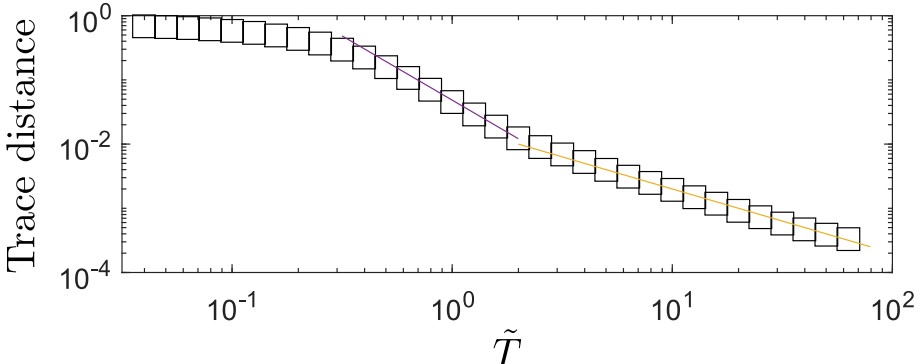

Figure 4: Trace distance between the exact equilibrium state of the quantum rotor and the Gibbs state (31) as a function of the dimensionless temperature $\tilde{T}$ for $\tilde{\hbar} = 1$ ($V_0 = \hbar^2/I$). To guide the eye, we also mark the slope $\tilde{T}^{-2} \propto \epsilon_2$ at intermediate temperatures, where the state is still affected by the potential, and the slope $\tilde{T}^{-1} \propto \epsilon_1$ at high temperatures in the quasi-free rotor regime. The exact equilibrium states are obtained by numerically evolving the Gibbs state for long times.

but also genuine quantum features such as tunneling out of a local potential minimum. This suggests a straight forward implementation for quantum tests and technological applications on mesoscopic scales. The relevance of our model is not restricted to experiments explicitly exploiting orientational degrees of freedom, but can also be used to estimate any rotational corrections in experiments addressing the center-of-mass degree of freedom.

The problem can be expanded to linear, symmetric or even asymmetric rotors, each for which the thermalization Lindbladian is formulated [31] and an external potential may be added via (trigonometric functions of) Euler angle operators to the Hamiltonian. While the analytical results shown here may very well be achieved in analogous fashion with existing phase space representations [53–55], each additional Euler angle adds a significant layer of complexity.

## Acknowledgments

Bi. S. and Bj. S. contributed equally. We thank Stefan Nimmrichter for valuable comments on our work and critical proofreading. We also thank Klaus Hornberger and Benjamin A. Stickler for stimulating discussions.

**Funding information**    This work was supported by Deutsche Forschungsgemeinschaft (DFG–449674892 and DFG–394398290).

## A   Steady state

In the classical high-temperature limit, the Gibbs state $\rho_G \sim e^{-\hat{H}/k_B T}$ is, up to first-order corrections in the small parameters $\epsilon_{1,2}$, the steady state of the dissipator (16). To show this explicitly, we start from the expression (19) for the dissipator acting on the Gibbs state, $\mathcal{L}\rho_G$. If we omit the potential energy for the moment, the series of commutator terms in (21) be-

comes a power series in $\epsilon_1 = \hbar^2/k_B T I \to 0$,

$$F(\hat{B})\big|_{V=0} = \sum_{k=0}^{\infty} \frac{(-\epsilon_1)^k}{2^k k!} \left[\frac{\hat{p}_\alpha^2}{\hbar^2}, \hat{B}\right]_k. \tag{A.1}$$

Combined with the $1/\epsilon_1$ prefactor in (19), we see that the summands with $k \geq 2$ contribute to the first-order correction in $\epsilon_1$ to the steady state condition. The lower-order terms in the remaining part cancel,

$$
\begin{aligned}
\mathcal{L}\rho_{\mathrm{G}}\big|_{V=0} &= \frac{2\Gamma}{\epsilon_1}\left\{\hat{A}\cdot\sum_{k=0}^{1}\frac{(-\epsilon_1)^k}{2^k k!}\left[\frac{\hat{p}_\alpha^2}{\hbar^2}, \hat{A}^\dagger\right]_k - \hat{A}^\dagger\cdot\hat{A}\right\}\rho_{\mathrm{G}} + \mathcal{O}(\epsilon_1) \\
&= \frac{2\Gamma}{\epsilon_1}\left[\left(\mathbf{e}_r(\hat{\alpha}) + \frac{i\epsilon_1}{4\hbar}\mathbf{e}_\varphi(\hat{\alpha})\hat{p}_\alpha\right)\cdot\left(\mathbf{e}_r(\hat{\alpha}) - \frac{i\epsilon_1}{4\hbar}\hat{p}_\alpha\mathbf{e}_\varphi(\hat{\alpha}) - \frac{\epsilon_1}{2}\mathbf{e}_r(\hat{\alpha})\right) - 1\right]\rho_{\mathrm{G}} + \mathcal{O}(\epsilon_1) \\
&= \frac{2\Gamma}{\epsilon_1}\left[\left(1 + \frac{\epsilon_1}{4} + \frac{\epsilon_1}{4} - \frac{\epsilon_1}{2}\right)|\mathbf{e}_r(\hat{\alpha})|^2 - 1\right]\rho_{\mathrm{G}} + \mathcal{O}(\epsilon_1) = \mathcal{O}(\epsilon_1).
\end{aligned} \tag{A.2}
$$

Here we have used that $|\mathbf{e}_r|^2 = 1$, whereas $\mathbf{e}_r \cdot \mathbf{e}_\varphi = 0$.

If we now re-insert the potential energy term $\hat{V}$ into the Hamiltonian $\hat{H}$, we must take into account mixed commutator terms up to $k = 3$ with one occurrence of $\hat{V}$ in (21), contributing in leading order $\hbar^2 V''(\alpha)/k_B^2 T^2 I \leq \epsilon_2$, to show that the steady state is also shaped by an appreciably strong potential. It turns out that all terms $\propto \hbar^2 V'(\alpha)/k_B^2 T^2 I$ with contributions of the potential $V(\alpha)$ vanish after operator re-ordering. For better readability we will only explicitly show terms including $V'(\hat{\alpha})$ leading to

$$
\begin{aligned}
\mathcal{L}\rho_{\mathrm{G}} &= \frac{2\Gamma k_B T I}{\hbar^2}\left(\hat{A}\cdot\left(-\frac{1}{k_B T}[\hat{V}, \hat{A}^\dagger] + \frac{1}{2k_B^2 T^2}[\hat{H}, \hat{A}^\dagger]_2\right) - \frac{(-k_B T)^{-k}}{k!}[\hat{V}, \hat{A}^\dagger\cdot\hat{A}]\right)\rho_{\mathrm{G}} + \mathcal{O}(\epsilon_1, \epsilon_2^2) \\
&= \frac{i\hbar\Gamma}{16 k_B^2 T^2 I}\Big[-2\hat{p}_\alpha V'(\hat{\alpha}) + 12\hat{p}_\alpha V'(\hat{\alpha}) + 4V'(\hat{\alpha})\hat{p}_\alpha \\
&\qquad - \frac{1}{3}\left(28\hat{p}_\alpha V'(\hat{\alpha}) + 20 V'(\hat{\alpha})\hat{p}_\alpha\right) + \hat{p}_\alpha V'(\hat{\alpha}) + V'(\hat{\alpha})\hat{p}_\alpha\Big]\rho_{\mathrm{G}} + \mathcal{O}(\epsilon_1, \epsilon_2^2).
\end{aligned} \tag{A.3}
$$

Higher order derivatives of $V(\alpha)$ are also included in $\mathcal{O}(\epsilon_2^2)$. In the second line, a systematic momentum operator ordering is not yet applied; doing this cancels almost all terms in the square bracket, except for the commutator

$$
\begin{aligned}
\mathcal{L}\rho_{\mathrm{G}} &= \frac{i\hbar\Gamma}{k_B^2 T^2 I}[\hat{p}_\alpha, V'(\hat{\alpha})]\rho_{\mathrm{G}} + \mathcal{O}(\epsilon_1, \epsilon_2^2) \\
&= \frac{\hbar^2\Gamma}{k_B^2 T^2 I}V''(\hat{\alpha})\rho_{\mathrm{G}} + \mathcal{O}(\epsilon_1, \epsilon_2^2) = \mathcal{O}(\epsilon_1, \epsilon_2).
\end{aligned} \tag{A.4}
$$

The relation $\epsilon_2 \propto \epsilon_1 V_0/k_B T$ highlights the fact that, even if the free rotor would thermalize very close to the Gibbs state, the potential could still be large enough so that $\epsilon_2 \gtrsim 1$ despite $\epsilon_1 \ll 1$. This would correspond to a deep quantum regime in which the level spacing of $\hat{H}$ is dominated by the potential energy, and it would no longer be justified to omit higher-order $\epsilon_2$-terms in (A.3).

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
