# Peer review of "Thermalization of the Quantum Planar Rotor with external potential"

_SciPost Physics Core, doi:SciPost Phys. Core 8, 040 (2025)_

## Round 2 · Referee Report · Anonymous (Referee 1) · 2024-12-17

Strengths

1 important problem addressed
2 novel & timely
3 clearly written

Weaknesses

1

Report

The manuscript by Schrinski et al. studies the thermalization dynamics of a planar quantum rotor in the presence of an external potential. The authors present a Wigner-Weyl formalism suitable for periodic degrees of freedom, which is then used to numerically show that the system relaxes into a steady state. These numerical calculations are supplemented by analytical results, discussing how the corresponding steady state approaches the Gibbs state for high temperatures.

The manuscript addresses the important open problem of how non-linear but periodic degrees of freedom thermalize. The manuscript is novel and timely, and well written. I thus recommend publication of the work in SciPost Physics Core after the authors considered the following minor comments on the presentation of their work.

  1. Though the main focus on the work is on thermalization of planar rotors, the corresponding dissipator is not introduced before Eq. (16) on page 4. I do acknowledge that the Wigner Weyl formalism for the planar rotor is non-trivial and central to this work, but it has been published before. I am thus wondering if Sec. II (apart from a few definitions) could be moved to the Appendix to streamline the manuscript on what is novel.

  2. At the beginning of Sec. V the authors introduce the dimensionless time, temperature, and Planck constant. While this might be suitable for the numerical implementation, the readability of the subsequent discussion would profit from explicitly writing out the expressions. For instance in and around Eq. (3), or in Figs. 3 and 4.

  3. It would be great if the authors could briefly comment in the conclusion on how their methods could be extended to 3D rotors and what challenges they expect there.

Requested changes

The above comments are optional

Recommendation

Publish (easily meets expectations and criteria for this Journal; among top 50%)

  • validity: top
  • significance: high
  • originality: high
  • clarity: high
  • formatting: perfect
  • grammar: perfect

Author:  Björn Schrinski  on 2025-03-04  [id 5263]

(in reply to Report 1 on 2024-12-17)

The Referee wrote: The manuscript by Schrinski et al. studies the thermalization dynamics of a planar quantum rotor in the presence of an external potential. The authors present a Wigner-Weyl formalism suitable for periodic degrees of freedom, which is then used to numerically show that the system relaxes into a steady state. These numerical calculations are supplemented by analytical results, discussing how the corresponding steady state approaches the Gibbs state for high temperatures. The manuscript addresses the important open problem of how non-linear but periodic degrees of freedom thermalize. The manuscript is novel and timely, and well written. I thus recommend publication of the work in SciPost Physics Core after the authors considered the following minor comments on the presentation of their work. 1. Though the main focus on the work is on thermalization of planar rotors, the corresponding dissipator is not introduced before Eq. (16) on page 4. I do acknowledge that the Wigner Weyl formalism for the planar rotor is non-trivial and central to this work, but it has been published before. I am thus wondering if Sec. II (apart from a few definitions) could be moved to the Appendix to streamline the manuscript on what is novel. Our answer: We confirm that the formulas and concepts before Eq. 18 are not original, a fact we tried to acknowledge by proper citation. However, we think that our concise recap of the topic is beneficial for the typical reader, even for experts in the field of quantum rotors. As suggested by the other referees we now explicitly declare which parts of the manuscript are truly novel. The Referee wrote: 2. At the beginning of Sec. V the authors introduce the dimensionless time, temperature, and Planck constant. While this might be suitable for the numerical implementation, the readability of the subsequent discussion would profit from explicitly writing out the expressions. For instance in and around Eq. (3), or in Figs. 3 and 4. Our answer: We added explicit expressions at suitable instances throughout section V. The Referee wrote: 3. It would be great if the authors could briefly comment in the conclusion on how their methods could be extended to 3D rotors and what challenges they expect there. Our answer: We are positive that our approach can be readily applied to linear and (a-)symmetric rotors, but the increase of complexity in the calculations is significant. We added a statement to the conclusion.

---

## Round 2 · Referee Report · Anonymous (Referee 2) · 2025-1-20

Strengths

1)Nice theoretical/analytical results

2)Hard problem

Weaknesses

1)Analytical result not very rigorous

2)Relation with previous literature not very clear

Report

The authors study how the thermalisation Lindblad operator behaves at large temperatures for 1D planar rotors. In particular, they proof analytically that the Gibbs state at large temperature is a steady states for this Lindblad operator.

I find the result interesting, and the derivation elegant. Moreover the problem of quantizing rotors is hard, hence every novelty in this sense is more than welcome. I suggest publication with minor revisions.

Requested changes

Here are some comments to be addressed:

1) Although quite elegant, the analytical derivation of the high temperature limit on Gibbs state is not very rigorous. Mathematically, it is not truly justified to develop in exponential power series the operator exp(H) when H is unbounded as in the case of the rotational kinetic energy operator. I understand that this is an intuitive physical explanation, but some comments on the validity/convergence of the series (21) (or more precisely (A1)) would be welcome.

2)It is not clear to me if there are any other previous analytical contributions on the high temperature limit of quantised rotors, hence it would be important to point this out more clearly.

Recommendation

Publish (easily meets expectations and criteria for this Journal; among top 50%)

  • validity: good
  • significance: good
  • originality: good
  • clarity: good
  • formatting: good
  • grammar: good

Author:  Björn Schrinski  on 2025-03-04  [id 5264]

(in reply to Report 2 on 2025-01-20)

The Referee wrote: The authors study how the thermalisation Lindblad operator behaves at large temperatures for 1D planar rotors. In particular, they proof analytically that the Gibbs state at large temperature is a steady states for this Lindblad operator. I find the result interesting, and the derivation elegant. Moreover the problem of quantizing rotors is hard, hence every novelty in this sense is more than welcome. I suggest publication with minor revisions. Requested changes Here are some comments to be addressed:

1) Although quite elegant, the analytical derivation of the high temperature limit on Gibbs state is not very rigorous. Mathematically, it is not truly justified to develop in exponential power series the operator exp(H) when H is unbounded as in the case of the rotational kinetic energy operator. I understand that this is an intuitive physical explanation, but some comments on the validity/convergence of the series (21) (or more precisely (A1)) would be welcome. Our answer: The referee is correct that in general the series (A1) does not converge because of the unbound operators involved. The referee is also correct that a rigorous mathematical proof would require a proper handling of the exponential map (with involved Lie Algebra techniques). Alternatively, in true physicists fashion and analogous to the numerical calculations, we truncate the Hilbert space at a sufficiently high angular momentum which allows us to use the simple form of the Baker-Campbell-Hausdorff decomposition shown. Any emerging errors are exponentially suppressed when eventually applying the Gibbs state. We added a comment (Ref. [55]) on this matter.

The Referee wrote: 2)It is not clear to me if there are any other previous analytical contributions on the high temperature limit of quantised rotors, hence it would be important to point this out more clearly. Our answer: There is a large body of work about quantized nano-rotors theory developed in the groups of Hornberger, Kim, Robicheaux, Romero-Isart, Stickler and others. We tried to acknowledge the work on decoherence, diffusion and thermalization in the introduction with (updated) references. Our work here is an extension of Ref. [31] with new results for non-inversion symmetric rotors and external potentials, also giving a small didactic introduction to the problem. Other work specifically on thermalization of planar rotors we are aware of are Refs. [33,34] based on approximative numerics.

---

## Round 2 · Referee Report · Anonymous (Referee 3) · 2025-1-21

Strengths

  1. didactically well written and clearly presented study
  2. timely

Weaknesses

  1. not entirly clear which part of the derivation are original
  2. relation to physical systems in terms of paramters (temperature, time, friction rate, depth of the potential) in physical units is lacking

Report

The authors present a formalism to describe the thermalization of a planar rotor in the presence of an external potential, using the Weyl-Wigner formalism and derive analytical as well as numerical solutions. The recent interest in levitating rotors on the mesoscopic/nano scale makes this a very timely study. The manuscript is well written, and the complex mathematical formalism is clearly and didactically well presented.

In my opinion, it is appealing to start introducing the Wigner-Weyl formalism with the well-known Cartesian case and then work out the differences that occur in the periodic phase space. The drawback of this didactical approach is that it is not entirely clear from which point on the derivation presented here is original – I suggest that the authors state this more clearly.

What I was missing in this manuscript, is the relation of the formal results to physical systems. For example, in which situations would I expect a (nano scale) rotor to be in the low or high temperature limit? On which timescale would such a thermalization take place? Are typical values for the parameters as the friction rate known?
I was also a bit puzzled by the choice of the initial state to be located at the maxima of the external potential – is there a particular setup the authors have in mind that is described by such initial state?

I assume that to work out a complete working physical example is beyond the scope of this study, but at least an estimate of the relevant numbers and their relation to real physical systems would greatly help the reader to access the relevance of the results.

Requested changes

  1. state more clearly which part of the derivation is original

  2. Give estimates for the relevant parameters in physical units, set results in context of physical systems (details see report)

Recommendation

Ask for minor revision

  • validity: -
  • significance: good
  • originality: good
  • clarity: high
  • formatting: excellent
  • grammar: -

Author:  Björn Schrinski  on 2025-03-04  [id 5265]

(in reply to Report 3 on 2025-01-21)

The Referee wrote: The authors present a formalism to describe the thermalization of a planar rotor in the presence of an external potential, using the Weyl-Wigner formalism and derive analytical as well as numerical solutions. The recent interest in levitating rotors on the mesoscopic/nano scale makes this a very timely study. The manuscript is well written, and the complex mathematical formalism is clearly and didactically well presented.
In my opinion, it is appealing to start introducing the Wigner-Weyl formalism with the well-known Cartesian case and then work out the differences that occur in the periodic phase space. The drawback of this didactical approach is that it is not entirely clear from which point on the derivation presented here is original – I suggest that the authors state this more clearly.
Our answer: We now explicitly state at the beginning of the introducing chapter that it is not original work. Original results start with formula (18) which we now also state in the text.

The Referee wrote: What I was missing in this manuscript, is the relation of the formal results to physical systems. For example, in which situations would I expect a (nano scale) rotor to be in the low or high temperature limit? On which timescale would such a thermalization take place? Are typical values for the parameters as the friction rate known?
Our answer: We now report typical numbers for state of the art experiments and proposals with nano-rotors at the end of chapter III. We also briefly discuss how the mean thermal occupation number depends on temperature and moment of inertia and how the low temperature limit may be reached.
The Referee wrote: I was also a bit puzzled by the choice of the initial state to be located at the maxima of the external potential – is there a particular setup the authors have in mind that is described by such initial state?
Our answer: The first state is located in the local minimum to show the tunneling effect during the thermalization process. Such escape process studies are widespread in open-system literature, quantum as well as classical, and we deemed it fitting here. The second example is supposed to mainly show the fast decoherence and is chosen to be localized somewhere else for maximum variation in the subsequent thermalization process as compared to the other example.
Both initial states are far fetched. A realistic setup for a first experimental realization is a rotor cooled to the deep quantum regime of a cos(2*alpha) potential (dielectric rotor in a standing wave). We now state in the second paragraph of chapter V that the shown examples are hard to prepare and are rather chosen to show as much physics as possible and the full capacity of Eq. (18). We discuss this in the second paragraph of chapter V.
The Referee wrote: I assume that to work out a complete working physical example is beyond the scope of this study, but at least an estimate of the relevant numbers and their relation to real physical systems would greatly help the reader to access the relevance of the results.
Our answer: We now give experimental numbers for typical (proposed) quantum experiments with nano-rotors at the end of section III.
Requested changes
The Referee wrote: 1. state more clearly which part of the derivation is original
Our answer: We adapted the text as explained above to make it more clear.

The Referee wrote: 2. Give estimates for the relevant parameters in physical units, set results in context of physical systems (details see report)
Our answer: A paragraph and a table are included on page 5 reporting realistic parameters.

---

## Round 3 · Referee Report · Anonymous (Referee 1) · 2025-3-9

Report

I thank the authors for their reply. I recommend publishing the paper in its current form.

Recommendation

Publish (easily meets expectations and criteria for this Journal; among top 50%)

---

## Round 3 · Referee Report · Anonymous (Referee 2) · 2025-3-31

Report

The authors successfully replied to my comments.

Recommendation

Publish (easily meets expectations and criteria for this Journal; among top 50%)

---

## Round 3 · Referee Report · Anonymous (Referee 3) · 2025-4-8

Report

The authors have addressed my comments satisfactorily; in particular, the more didactical part and the new results are now clearly identified and typical parameters for nano-rotors are given.
I therefore recommend publication of the maunscript.

Recommendation

Publish (easily meets expectations and criteria for this Journal; among top 50%)

---

## Round 3 · List of Changes

• We added statements about the degree of originality in the different parts of the manuscript.
  • The dimensionless parameters are expressed explicitly at suitable instances.
  • Typical experimental parameters for (proposed) experimental realizations with levitated nano-rotors are reported.
  • The list of references has been updated with several citations.

---

## Editorial Decision

published